# Developing a Theory-Based Instrument for Pre-Exposure Prophylaxis (PrEP) Uptake in People of Color Using a Qualitative Approach

**DOI:** 10.3390/healthcare12161595

**Published:** 2024-08-10

**Authors:** Siddharth Raich, Christopher Johansen, Neeraj Bhandari, Kavita Batra, Manoj Sharma

**Affiliations:** 1Department of Social and Behavioral Health, School of Public Health, University of Nevada, Las Vegas, NV 89119, USA; christopher.johansen@unlv.edu (C.J.); manoj.sharma@unlv.edu (M.S.); 2Department of Healthcare Administration and Policy, School of Public Health, University of Nevada, Las Vegas, NV 89119, USA; neeraj.bhandari@unlv.edu; 3Department of Medical Education and Office of Research, Kirk Kerkorian School of Medicine, University of Nevada, Las Vegas, NV 89102, USA; kavita.batra@unlv.edu; 4Department of Internal Medicine, Kirk Kerkorian School of Medicine, University of Nevada, Las Vegas, NV 89102, USA

**Keywords:** Pre-exposure Prophylaxis (PrEP), communities of color, Multi-theory Model

## Abstract

There is a large disparity in Pre-exposure Prophylaxis (PrEP) utilization among communities of color compared to White Americans. There is also a lack of theory-based survey instruments to measure the underlying reasons for the disparity among communities of color. The purpose of this study was to create an instrument based on a qualitative approach involving community interviews. Semi-structured interviews guided by the Multi-theory Model (MTM) of health behavior change were performed in a sample of 12 members from communities of color. The analysis entailed a directed content analysis along the themes of MTM constructs to develop a survey instrument. The barriers to PrEP that emerged included the cost of PrEP, lack of protection from other sexually transmitted diseases, reduced trust between partners, and the stigma associated with PrEP. The perceived disadvantages included the potential cost of PrEP, partner mistrust when taking PrEP, discussion of sexual behaviors with a provider, and unclear process of acquiring the PrEP prescription. The results guided the development of a survey tool to further investigate aspects of cost, partner relations, stigma, reassurance of safety, and other factors. The tool can be used for future studies as part of guided interventions to increase PrEP uptake.

## 1. Introduction

The breakthrough measure of Pre-Exposure Prophylaxis (PrEP) in HIV/AIDS prevention was approved by the FDA in 2012 and ultimately added to the regimen advocated by the World Health Organization (WHO) [1]. When used as intended, PrEP can prevent new HIV infections by nearly 99% [2]. PrEP can be consumed orally through a once-daily tablet, or intravenously through a monthly injection [2].

Although PrEP is highly effective in the prevention of HIV, it is not utilized equally by every racial or ethnic group. There is a large disparity in Pre-exposure Prophylaxis (PrEP) utilization among communities of color as compared to White Americans in the United States with nearly 9% of eligible members of communities of color using PrEP compared to 66% of eligible White Americans using PrEP [3]. This research investigates possible reasons for the discrepancies in PrEP usage and the factors that influence its availability to inform the development of the survey tool. Racial and ethnic minorities, LGBTQIA+ individuals, persons from lower socioeconomic levels, and individuals residing in rural locations may have difficulties in receiving PrEP services [4]. Cultural considerations, such as the negative social perception associated with HIV and sexuality, may discourage people from accessing PrEP programs, especially in conservative cultures [5]. Although there is a strong urgency to develop customized treatments to tackle PrEP inequities, there is a significant absence of theory-based tools, particularly for communities of color in the field of public health [6]. The use of theory-based instruments allows for a systematic approach to comprehending the factors that influence health behaviors and developing impactful interventions. However, current instruments frequently overlook the interconnected socioeconomic factors that influence the use of PrEP among people of color. In addition, numerous measures lack cultural sensitivity and relevance to varied communities, hence worsening inequities in healthcare access and outcomes. To address the inequalities in PrEP availability, it is necessary to implement comprehensive interventions based on the baseline data collected through robust survey instruments [7]. These comprehensive interventions may encompass community outreach and education programs, cultural competency training for healthcare personnel, policy reforms aimed at increasing PrEP coverage and reducing financial obstacles, and the creation of theory-based instruments specifically designed for minority communities [7]. Resolving inequalities in PrEP availability necessitates a collaborative endeavor involving politicians, healthcare professionals, community groups, and researchers. This research aims to fill the void of theory-based quantitative survey instruments to measure the underlying reasons for the disparity among communities of color. 

This research utilizes the Multi-theory Model (MTM), a fourth-generation theory of health behavior change that considers factors that influence initial behavioral change as well as factors that influence continued behavior [8,9]. MTM consists of two key concepts of *initiation* and *sustenance* comprising of three constructs each. The six constructs of MTM are participatory dialogue, behavioral confidence, changes in the physical environment, practice for change, emotional transformation, and changes in the social environment [8,9]. MTM has been used in a qualitative setting in numerous studies from short-term to long-term changes in health behavior [10,11]. MTM has also been utilized in cross-sectional studies including predicting physical activity in Chinese pregnant women [12]. As PrEP requires a prescription for initial use, followed by the sustenance of the once-daily tablet or monthly injections, the constructs of MTM are well suited for this research. The purpose of this research is to create a qualitative instrument structured with the Multi-theory Model (MTM) of health behavior based on a directed content analysis from qualitative interviews. 

## 2. Materials and Methods

### 2.1. Study Design

The design of the study was qualitative using a critical theory paradigm with in-depth interviews based on a predetermined framework of the multi-theory model (MTM) of health behavior change [8]. Directed content analysis was chosen for this research using NVivo software to create codes for each respondent based on theoretical constructs. Directed content analysis is a summative approach for data analysis consisting of a step-by-step process to obtain valuable information from interviews [13]. Directed content analysis also assists researchers in comparing different studies by yielding practical results through transparent themes [13]. The codes surrounding the a priori themes of perceived advantages and disadvantages, behavioral confidence, changes in the physical environment, practice for change, emotional transformation, and changes in the social environment from each respondent were ultimately collated and led to the creation of overarching themes in respondent answers to help guide instrumentation development.

An interview protocol was designed using the fourth-generation Multi-Theory Model (MTM) of health behavior change to explain the initiation and maintenance of PrEP usage. Appendix A provides details of the protocol and the instrument. The inclusion criteria were the following: participants must be over the age of 18, sexually active, HIV-negative, not currently on PrEP, in a polyamorous relationship or single and dating multiple partners, and someone who identifies as a person of color. The exclusion criteria were the following: any participants under the age of 18, not sexually active, currently on PrEP, has ever had an HIV-positive diagnosis, and in a monogamous relationship. 

### 2.2. Recruitment Methodology

The methodology for recruitment included the posting of electronic flyers on open forums and public pages that were accessible to the public. Websites also included pages for gender and sexual minorities and people of color. Participants were able to enroll by either calling the researcher via telephone or following a QR-coded link to sign up. Once participants desired to sign up, they were guided to a screening survey that asked questions based on the inclusion and exclusion criteria to ensure eligibility. 

Eligible participants were taken to a webpage to collect contact information and eligible dates and times for their convenience were gathered. The interviewer then informed the eligible participants that an interview could be conducted electronically via Zoom or another web-based platform, or in person at a reserved conference room on a large Southwestern University campus. All participants opted to interview electronically. All participant information was stored on password-protected computers using password-protected documents. 

### 2.3. Data Analysis

The data analysis for this phase comprised directed content analysis using NVivo 14.0 software to create themes for each respondent based on theoretical constructs. First, themes were created based on the MTM constructs of perceived advantages and disadvantages, behavioral confidence, changes in the physical environment, practice for change, emotional transformation, and changes in the social environment. Subsequently, responses from each participant were collated and categorized based on overarching themes to help guide quantitative instrumentation development [8]. Lastly, the contents of each theme were analyzed, and questions were developed based on the respective information for each theme. 

After the instrument was finalized with the help of qualitative methods it was validated by a panel of six experts in two rounds. There were three subject matter experts, five target population experts, three instrumentation experts, and four multi-theory model (MTM) framework experts. Expertise in multiple areas was shared by some experts. The first round resulted in 11 changes in the instrument for improvement in readability and the second round yielded 4 changes for a total of 15 changes from the panel of experts. 

## 3. Results

The participants’ ages (in years) were as follows: 27 (Participant #1), 34 (Participant #2), 21 (Participant #3), 23 (Participant #4), 37 (Participant #5), 29 (Participant #6), 22 (Participant #7), 21 (Participant #8), 36 (Participant #9), 41 (Participant #10), 22 (Participant #11), and 22 (Participant #12). The mean age was 27.92 (standard deviation + 7.28). Three participants were African American (25%), six were Latinx (50%), and three respondents were Asian (25%). Eleven out of 12 participants (92%) identified as being part of a sexual minority, and one respondent stated they were unsure about their sexual orientation. There was a total of nine men (75%) and three women (25%) that comprised the interview pool. All 12 respondents stated they had health insurance. Lastly, 10 out of the 12 respondents (83.3%) stated that they did currently practice safe sex practices, such as the use of condoms, dental dams, and IUDs, while 2 did not (16.7%). Table 1 below highlights the demographics and information of participants.

### 3.1. Interview Results

The sample size (n = 12) of interviews led to ample qualitative data based on participant beliefs, attitudes, knowledge, and experiences. Only 3 of the 12 participants (25%) mentioned that they had heard of PrEP, and the other 9 participants (75%) were provided a brief overview of PrEP. 

### 3.2. Initiation

#### 3.2.1. Perceived Advantages of PrEP

In terms of advantages of PrEP, recurring themes included “peace of mind” or “mental peace” (Survey Respondent: 3, 6, 9, 11, 12), “protection from HIV” (Survey Respondents: 1, 3, 4, 5, 6, 7, 10), “one pill a day” (Survey Respondents: 1, 2, 8, 11, 12), and the need for fewer “HIV check-ups” (Survey Respondents: 2, 4, 7, 10). The first advantage of peace of mind was summarized by respondent 3 as *“If I take PrEP I would have the peace of mind that I wouldn’t get HIV*”. Other respondents also mentioned that they would feel a state of “mental peace” and not have to worry about acquiring HIV. The second advantage was appropriately the “protection from HIV” as respondents were informed that PrEP is highly effective in protecting from HIV when used as directed. The third advantage was that it was easy to consume, with numerous respondents highlighting that it simply consisted of taking one pill each day. The final advantage that was stated by multiple respondents was the need for fewer HIV check-ups, as many stated that HIV checkups are a recurring part of life for individuals in sexual minorities. 

*The perceived advantages* construct was defined as the potential benefits of an action as imagined by the respondent. In this study, the advantages of PrEP are operationalized by attributes identified in the qualitative study including peace of mind, personal protection against the uptake of HIV/AIDS, and the protection of sexual partners. The items in the instrument for perceived advantages include the following: If you use PrEP, you will have greater peace of mind; taking PrEP is easy for you; if you use PrEP, it will eliminate the need for regular HIV check-ups; and if you use PrEP, you will be protected from getting HIV. They were measured on a summative scale of Not at all likely (0), Hardly likely (1), Moderately likely (2), Very likely (3), and Completely likely (4). The score range for this construct is 0–16. 

#### 3.2.2. Perceived Disadvantages of PrEP

Several disadvantages of PrEP also surfaced from the qualitative surveys with themes of potential partner “mistrust” (Survey Respondents: 4, 8, 9, 10), handling the stigma and “people judging” participants for taking PrEP (Survey Respondents: 1, 4, 5, 8, 9, 10, 12), lack of protection from sexually transmitted infections (STIs) (Survey Respondents: 2, 4, 11, 12), and the potential cost of PrEP (Survey Respondents: 3, 7, 9, 10). The theme of potential partner mistrust was stated by Respondent 4 as *“My boyfriend would think that I don’t trust him”* and Respondent 8 as *“My partner would think that I think he’s lying about not having HIV”.* Another theme that emerged was facing the stigma and *“people might find out that I’m taking PrEP”* (Participant 4). Survey respondent 2 emphasized that PrEP only protects against HIV and not *“other STIs, I’m more worried about [other] STIs than I am about HIV”* as a potential disadvantage of taking PrEP. Three other respondents also elaborated on the fact that PrEP only protects from HIV and not STIs such as herpes, chlamydia, and gonorrhea, etc. Lastly, the recurring theme of cost was discussed by respondents 3, 7, 9, and 10, with all of them describing that PrEP may be “expensive”. Perceived *Disadvantages* were defined as the perceived harm or drawback of an action. In this study, the disadvantages were operationalized by attributes identified in the qualitative study including the ineffectiveness of PrEP, partner mistrust, may not be readily available, and others may discover participant PrEP use. The instrument included the following questions: You may not be able to afford to use PrEP if it costs as much as most other prescription medications; regular use of PrEP may not protect you against other sexually transmitted infections; use of PrEP may lead to reduced trust with your sexual partner(s); and the use of PrEP may lead people to judge you. They were measured on a summative scale of Not at all likely (0), Hardly likely (1), Moderately likely (2), Very likely (3), and Completely likely (4). 

Based on these themes, the constitutive definition of the construct of *Participatory Dialogue* was the component of the multi-theory model (MTM) of health behavior relating to the initiation construct defined as the dialogue between health educators and study participants [8,9]. The term participatory dialogue was operationally defined as the advantages of PrEP and the disadvantages of PrEP and measured by subtracting the disadvantage score from the advantage score to create a final score. The score range for this construct is −16 to +16 as in previous MTM studies [14].

#### 3.2.3. Behavioral Confidence

When participants were asked about building behavioral confidence key themes included support of their partner (Survey Respondents: 3, 4, 7, 9, 10), reminders to take the pill (Survey Respondents: 1, 2, 7, 11), building self-assurance (Survey Respondents: 6, 10, 11), and doing some more research (Survey Respondents: 8, 9). Survey respondent 3 highlighted that *“my confidence to take PrEP is going to happen if my partner supports me or if they start to take PrEP every day too”.* Furthermore, respondent 7 stated that *“I’ll be more confident to take it if my boyfriend takes it too”* emphasizing that the behavioral confidence may be strengthened by the support of their partner. Respondent 11 stated that they would be confident to take PrEP if they had “a daily reminder like an app or something” to build their confidence in consistently taking PrEP. Respondents 6, 10, and 11 all pointed towards qualities of self-assurance, with respondent 10 stating that they would exhibit more behavioral confidence to take PrEP if they believed that they could “make their own decisions and stick to it” to be protected from HIV. Respondents 8 and 9 also expressed that their behavioral confidence would increase if they conducted their own research into PrEP and had a more in-depth understanding of the medication. *Behavioral Confidence is* the component derived from the initiation construct of the Multi-theory Model of health behavior defined as the confidence or belief that an individual can begin a particular health behavior change [8,9]. In this study, behavioral confidence was operationalized by attributes identified in the qualitative study including the confidence to get a PrEP prescription and ultimately begin PrEP usage. The items used in the instrument are How sure are you that you can start using PrEP soon if there are no barriers? How sure are you that you can start using PrEP even if it has a high cost? How sure are you that you can start using PrEP even if it decreases trust with your partner(s)? and How sure are you that you can start using PrEP despite being judged by others? The items were measured on a summative scale of Not at all sure (0), Hardly sure (1), Moderately sure (2), Very sure (3), and Completely sure (4). The score range for this construct is 0–16. 

#### 3.2.4. Changes in the Physical Environment

Several key themes arose during the discussion of changes in the physical environment. Respondents most notably stated that they would need to physically go to the doctor’s office (Survey Respondents: 3, 4, 5, 6, 7, 8, 10, 11). Survey respondents 5 and 6 also stated that they would need to “go to clubs less” and “not go to bars all the time” as changes in the physical environment, respectively. Survey respondents 2 and 3 emphasized the need to visit their local pharmacy to get the medication. The *changes in the physical environment* construct is the component derived from the initiation construct of the multi-theory model of health behavior change and is defined as the physical surroundings that promote the initiation of a health behavior change [8,9]. In this study, changes in the physical environment were operationalized by attributes identified in the qualitative study such as the access to obtain the prescription for PrEP, scheduling an appointment, the cost to transport to a healthcare provider, and transportation to fill prescriptions. The questions in the instrument include the following: How sure are you that you will be able to get to your doctor’s office for an appointment to get PrEP? How sure are you that you will be able to get PrEP through your local pharmacy? and How sure are you that you will be able to afford transportation if necessary to get PrEP? The items were measured on a summative scale of Not at all sure (0), Hardly sure (1), Moderately sure (2), Very sure (3), and Completely sure (4). The score range for this construct is 0–12. 

### 3.3. Sustenance

#### 3.3.1. Emotional Transformation

Emotions when taking PrEP were largely positive from survey respondents. Respondent 4 stated *“I would feel happy and safe knowing that I don’t have to stress out anymore”*, and similarly, respondent 7 stated that they *“would be glad there is one less thing to worry about”*. When asked about how they may change their emotion to achieve the goal of taking PrEP, respondent 6 stated that they would *“have to keep reminding myself that this pill is really good for me”.* Lastly, respondent 12 mentioned that they would feel “safe, knowing that me and my partners won’t get HIV”. *Emotional transformation is the* component derived from the sustenance construct of the Multi-theory Model of health behavior change defined as converting emotions towards a sustained health behavior change [8,9]. In this study, emotional transformation was operationalized by attributes identified in the qualitative study including directing emotions to set goals to continue daily PrEP intake or bi-monthly injections. The instrument questions include How sure are you that you can encourage yourself toward having one less worry of getting HIV? How sure are you that you can motivate yourself to use PrEP to protect yourself and your partner(s)? and How sure are you that you can overcome the fear of judgment from friends/relatives to accomplish the goal of using PrEP? The items were measured on a summative scale of Not at all sure (0), Hardly sure (1), Moderately sure (2), Very sure (3), and completely sure (4). The score range for this construct is 0–12. 

#### 3.3.2. Practice for Change

Approaches to employ in taking PrEP included regular “phone reminders” by respondents 5, 6, 9, and 10. Respondents 7 and 8 stated that taking PrEP along with their partners would serve as a regular reminder for both to take PrEP. Respondent 2 mentioned that “seeing their doctor” regularly would also serve as a reminder to continue the PrEP prescription and take PrEP. Respondent 11 mentioned “*I already take my birth control pill every day; I would take the PrEP at the same time if that’s safe*” as a method to combine PrEP with ongoing medication/pills. *Practice for change is* a construct of the Multi-theory Model of health behavior change defined as an individual’s thoughts about a health behavior change and a sustained adjustment of strategies and overcoming obstacles to ensure a long-term health behavior change [8,9]. In this study, practice for change was operationalized by attributes identified in the qualitative study including the ability to continue taking the daily pill for PrEP or to continue receiving bi-monthly injections for PrEP. Instrument items include “If you choose to take PrEP, phone reminders would be useful in helping you to keep taking PrEP”, and “If you choose to take PrEP you are sure that you can continue to take it even if your partner(s) are against it,” as well as “If you choose to take PrEP you are sure that you can continue taking it in the long run even if you forget to take it occasionally. Practice for change items were measured on a summative scale of Not at all sure (0), Hardly sure (1), Moderately sure (2), Very sure (3), and Completely sure (4). The score range for this construct is 0–12. 

#### 3.3.3. Changes in the Social Environment

The topic of social support raised numerous themes and elicited the most in-depth answers from survey respondents. When asked about the kind of social support that respondents would need for taking PrEP, one of the most notable responses was support from their “partners” (Survey Respondents: 1, 2, 3, 4, 6, 9, 11, 12). The interviews have indicated a great deal of connectedness with their partners for support, to instill confidence, to overcome barriers, and many other aspects of starting and sustaining PrEP. The second kind of support mentioned was support from friends. Respondent 7 emphasized that “*if my friends also start taking PrEP then we would all just take it together*”. Respondent 10 also stated that they would be “supported by my friends”. Respondent 5 stated that their “doctor would help me in getting PrEP”. Lastly, respondent 1 also stated that their mother would help support them in taking the once-daily pill. *Changes in the social environment* is the construct defined as the social support from the individual’s surroundings that creates a positive relationship towards continued behavioral change [8,9]. In this study, changes in the social environment was operationalized by attributes identified in the qualitative study including getting support from a sexual partner, family member, physician, or another trusted individual to ensure daily pill usage or bi-monthly injection of PrEP. Instrument items include: How sure are you that you can get the help of your doctor to support you in using PrEP? How sure are you that you can get the help of your partner to support you in using PrEP? and How sure are you that you can get the help of your friends in using PrEP? Changes in the social environment items were measured on a summative scale of Not at all sure (0), Hardly sure (1), Moderately sure (2), Very sure (3), and Completely sure (4). The score range for this construct is 0–12. 

The construct of the *intention of initiation* is the construct of the Multi-theory Model of health behavior change defined as a short-term change that transitions an individual to begin a particular health behavior [8,9]. In this study, the construct of intention of initiation was operationalized by attributes identified in the qualitative study including the likelihood of obtaining a prescription for PrEP soon, the desire to take PrEP soon, and the willingness to take PrEP soon. Instrument items include the following: How likely is it that you will use PrEP in the next month? How likely is it that you will want to use PrEP in the next month? and How likely is it that you intend to use PrEP in the next month? The construct items were measured on a summative scale of Not at all likely (0), Hardly likely (1), Moderately likely (2), Very likely (3), and Completely likely (4). The score range for this construct is 0–12. 

The construct of the *intention of sustenance* is the construct of the Multi-theory Model and is defined as the long-term change in health behavior [8,9]. In this study, the construct of the intention of sustenance was measured by attributes identified in the qualitative study such as the likelihood of continuing a prescription for PrEP, the desire to continue PrEP, and the willingness to continue PrEP. Instrument questions include the following: How likely is it that you will use PrEP from now on? How likely is it that you will use PrEP from now on? How likely is it that you will want to use PrEP from now on? and How likely is it that you intend to use PrEP from now on? Items will be measured on a summative scale of Not at all likely (0), Hardly likely (1), Moderately likely (2), Very likely (3), and Completely likely (4). The score range for this construct is 0–12. 

Table 2 below highlights the main results based on theoretical constructs and participant responses. 

Based on the results above as defined constitutively and operationally a quantitative instrument was created. The final instrument which was also validated in two rounds by the panel of experts, is provided in Appendix B.

### 3.4. Program Components

When asked about components of programs for people of color to help take PrEP, respondent answers can be summarized in three key themes: cultural sensitivity, programs led by people of color, and factual information about PrEP. Respondents 1, 6, 7, 9, 10, and 12 all stated that the program should be culturally sensitive and culturally appropriate for people of color. Respondents 2, 6, 7, and 11 also emphasized that programs for people of color should also be led by people of color. The respondents mentioned that this would build a certain level of trust in the program and the information provided. Many survey respondents mentioned that there is plenty of misinformation amongst people of color, especially among sexual minorities. Respondent 11 stated, “In the gay community, you hear so many things about new drugs and pills that prevent herpes and whatever, but half of the stuff is made up and they are passed around as true, so it’s hard to know what’s real and what’s made up”. Respondents 3, 4, and 8 also mentioned that the program should consist of factual and scientifically researched components to be better received by people of color and minority communities. In terms of the modality of delivery of such programs, most respondents preferred online sessions with two respondents stating they would prefer in-person education. 

## 4. Discussion

The research question for the qualitative phase of the study was as follows: What are the thoughts, perceptions, attitudes, and beliefs regarding PrEP for HIV prevention among people of color based on the theoretical paradigm of the multi-theory model (MTM) of health behavior change? The qualitative information gathered necessary information to adequately answer the research question. The qualitative information gathered from each respondent was diverse and full of anecdotes, beliefs, and experiences. As PrEP is a relatively new innovation, few qualitative studies have obtained theory-based information from people of color regarding PrEP. Willie et al. conducted a qualitative analysis of PrEP in 2021 and uncovered several themes including internal assets, responsibility, added protection, cost, mistrust, and side effects [15]. The results of this study are similar to the review of Willie et al. in that the topics of responsibility, protection, cost, and mistrust surfaced as themes throughout the interviews. However, this study included specific aspects regarding interpersonal relations with partners, consumption methods of PrEP, lack of open discussions with providers, the process of acquiring PrEP, and social support from family and partners. The instrument in Appendix B was developed based on qualitative feedback, and this section provides a rationale for the creation of quantitative instrument items. 

The perceived advantages of PrEP as highlighted in the results include having peace of mind. This mental health advantage of PrEP has not been thoroughly explored in previous studies regarding PrEP use amongst people of color. While many respondents also stated the physical protection from HIV as a notable advantage of PrEP, the mental peace of not having to worry about HIV appeared to be a prominent factor among sexual minorities within people of color. This finding was incorporated into the instrument with the item “If you use PrEP, you will have a greater peace of mind”. Respondents also stated that another advantage of taking PrEP was that it is simply one pill a day, rather than following complex instructions that vary over the course of the medication. The one pill a day provides a simple and efficient method of consuming PrEP and serves as a benefit when dealing with competing priorities. This feedback was incorporated into the survey with the item “Taking PrEP would be easy for you”. Lastly, the qualitative feedback showed that regular HIV check-ups are a large aspect of life among sexual minorities, and taking PrEP would eliminate the need for regular check-ups. This feedback was incorporated into the instrument with the item “If you use PrEP, it will eliminate the need for regular HIV check-ups”.

Numerous perceived disadvantages of taking PrEP also arose among the surveyed population. A notable disadvantage outlined in the results was the potential of partner mistrust about HIV status or promiscuity if one partner began taking PrEP. This result indicates the need for more guided dialogue training to encourage open communication with partners regarding the consumption of PrEP. An item was included in the instrument to reflect this feedback: “Use of PrEP may lead to reduced trust with your sexual partner(s)”. 

In alignment with previous research, using PrEP contains a degree of stigma. Cernasev et al. highlighted that there are numerous persistent barriers to the usage of PrEP [15]. An item was included in the instrument reflecting the feedback, “The use of PrEP may lead people to judge you”. Survey respondents also highlighted that a disadvantage of PrEP is the inability to protect from sexually transmitted infections (STIs). This feedback was incorporated into the item “Regular use of PrEP may not protect you against other sexually transmitted infections”. Current literature highlights that the cost of PrEP is often a barrier to the usage of PrEP [16]. This study also found that cost is a barrier for PrEP with many respondents believing that PrEP may be too “expensive” to use. An item was incorporated into the instrument stating “You may not be able to afford to use PrEP if it costs as much as most other medications”.

To obtain information on behavioral confidence, respondents were asked how they may build confidence in taking PrEP. The connection with the partner was a recurring aspect of PrEP usage for individuals who are part of sexual minorities. Previous research has also shown that reduced assurance from partners may impact the sustenance of health behaviors [15]. An item was developed stating “If you choose to take PrEP, you are sure that you can continue to take it even if your partner(s) are against it”. Individuals also stated that reminders to take the pill daily would help reduce lapses in taking PrEP. The item relating to reminders was “If you choose to take PrEP, phone reminders would be useful in helping you to keep taking PrEP”. Building self-assurance was also a theme to further reinforce the belief that they can sustain PrEP use. The question adapted based on this feedback was “If you choose to take PrEP, you are sure that you can continue taking it in the long run even if you forget to take it occasionally”. 

In terms of changes in the physical environment, doctor’s offices and pharmacies were appropriately mentioned as they are part of the process of PrEP uptake. Previous research has also shown that navigating the healthcare system is often a challenge for minority communities [17]. Therefore, items included in the instrument were: “How sure are you that you will be able to get to your doctor’s office for an appointment to get PrEP”? and “How sure are you that you will be able to get PrEP through your local pharmacy”? 

When respondents were asked about the emotions they would feel when taking PrEP, the emotions were largely optimistic. Items that were adapted for the instrument included the following: “How sure are you that you can encourage yourself toward having less worry of getting HIV”? and “How sure are you that you can motivate yourself to use PrEP to protect you and your partner(s)”? Another item regarding emotions included “How sure are you that you can overcome the fear of judgment from friends/relatives to accomplish the goal of using PrEP”?

The practice for change construct revolved largely around reminders. Previous research has also shown that the development of a routine or habit has a positive impact on PrEP behaviors [16]. Respondents mentioned phone reminders as an electronic method to serve as a daily reminder for the pill. Appropriately, a question was included in the instrument as “If you had daily phone reminders, would that assist you in regularly continuing PrEP”? Respondents also mentioned that seeing their doctor more frequently would serve as a reminder to continue PrEP. The question that was adapted from this feedback was “If you were to see your doctor more regularly, would you be more likely to continue PrEP usage”? Lastly, a female respondent associated PrEP usage with her birth control pills as a method of combining the uptake for both. A question adapted based on this feedback was “If you currently take daily birth control pills, would you be more likely to take PrEP pills as well”?

The construct of social support among this population was crucial to investigating beliefs and attitudes towards PrEP by people of color. Previous literature has also shown that the social structure plays a key role in PrEP usage [16]. Support from partners was highlighted yet again as a critical aspect of PrEP sustenance. The question adapted was “Would you be more likely to continue PrEP usage if you had the support of your partners”? Similarly, feedback regarding friends was incorporated by asking “Would you be more likely to continue PrEP usage if you had the support of your friends”? Lastly, the final question incorporated feedback about their doctor and asked, “Would you be more likely to continue PrEP if your doctor were to help you get PrEP”? 

Respondents were also asked for input regarding potential programs to increase PrEP usage among people of color. A question regarding cultural appropriateness was adapted as “Do you believe that a program should be culturally appropriate to increase PrEP usage among people of color”? Respondents also stated that programs should be led by people of color, leading to the question “Do you believe that programs led by people of color will be more effective in increasing PrEP usage among people of color”? As stated by numerous respondents, there are many myths and plenty of misinformation among people of color; therefore, having programs based on scientific research and facts would help improve their effectiveness. A question based on this feedback was incorporated as “If a program was scientifically grounded, would it better improve PrEP uptake among people of color”? 

The insights brought about by this research lead the way for future researchers to further investigate cultural, social, and structural obstacles that could impede access to PrEP, along with methods for surmounting them. Furthermore, analyzing the views, knowledge, and attitudes of PrEP within communities of color can provide valuable insights for developing focused educational campaigns and outreach initiatives. Public health practitioners can promote health equity and reduce HIV transmission rates by focusing research on the use of PrEP among communities of color.

## 5. Strengths and Limitations

### 5.1. Strengths 

Strengths of this research included the primary collection of qualitative data directly from the sample population of people of color. Data were also coded based on the Multi-theory Model to provide structure in coding and themes were created based on theoretical constructs. This allows for future replicability by researchers following the theoretical constructs as a guide for coding. The rigor of qualitative interviews until data saturation is a key strength of this study. The researcher first screened eligible participants, gathered consent, and then conducted an open semi-structured interview with each participant. The dependability of the study is high as the study procedures are transparent and results have been provided [17].

### 5.2. Limitations

The limitations include limited objectivity as it is often more difficult to record and demonstrate when compared with quantitative research [15]. Additionally, it is difficult to incorporate all feedback from each respondent in the study findings when compared to quantitative research [16]. This research was limited to communities of color and therefore did not include the views of White Americans which may include factors that lead to the disparity of PrEP use. The sample size of 12 serves as a limitation as it reduces the applicability of results to larger populations. Also, the study includes three females vs. nine males which may impact the results and represent the views of males more than females. Also, triangulation with quantitative methods has not yet been conducted and these findings are based solely on qualitative interviews. This also reduces the generalizability of this study as the qualitative interviews are regarding specifically PrEP uptake among communities of color [16]. The study includes three females vs. nine males which may impact the results as the sexual relationships among males and females vary. Similarly, if a study wanted to replicate these results quantitatively, there may be a concern regarding the unbalanced number of males and females.

Given the sensitive nature of topics regarding HIV and sexual behavior, desirability bias may be present in the study [15]. Respondents may answer questions based on the social acceptability of answers instead of their own experiences or beliefs [15]. Respondents may have in turn withheld compromising information about their sexual behavior and/or their beliefs about PrEP to conform with the perceived norm. Lastly, confirmability is difficult to obtain due to the nature of the qualitative research as complete neutrality was not possible. 

## 6. Conclusions

This research helped to develop an instrument using qualitative methods with a better understanding of the perceived advantages, perceived disadvantages, barriers, emotions, physical environment, and social environment of communities of color. The tool is guided by the Multi-theory Model and helps to gauge the intention to initiate PrEP use as well as the intention to sustain PrEP use among communities of color. The tool can be used for future studies and particular sub-populations as part of guided interventions to increase PrEP uptake and ultimately help reduce the disparity of new HIV infections.

## Figures and Tables

**Table 1 healthcare-12-01595-t001:** Participant Information.

**Summary Category**	**Data**
Age (s)	21, 21, 22, 22, 22, 23, 27, 29, 34, 36, 37, 41
Average Age	27.92 years
Ethnicity Distribution	
- African American	3 (25%)
- Latinx	6 (50%)
- Asian	3 (25%)
Sexual Orientation Distribution	
- Sexual Minority	11 (92%)
- Unsure	1 (8%)
Gender Distribution	
- Men	9 (75%)
- Women	3 (25%)
Health Insurance Coverage	12 (100%)
Safe Sex Practices	
- Practicing Safe Sex	10 (83.3%)
- Not Practicing Safe Sex	2 (16.7%)

**Table 2 healthcare-12-01595-t002:** Main Results.

Theoretical Construct	Respondent #
**Perceived Advantages of PrEP**	
Peace of mind/mental peace	3, 6, 9, 11, 12
Protection from HIV	1, 3, 4, 5, 6, 7, 10
One pill a day	1, 2, 8, 11, 12
Fewer HIV check-ups	2, 4, 7, 10
**Perceived Disadvantages of PrEP**	
Potential partner mistrust	4, 8, 9, 10
Stigma or judgment of others	1, 4, 5, 8, 9, 10, 12
Lack of protection from other STIs	2, 4, 11, 12
The potential cost of PrEP	3, 7, 9, 10
Behavioral Confidence	
Support of their partner	3, 4, 7, 9, 10
Reminders to take the pill	1, 2, 7, 11
Building self-assurance	6, 10, 11
Additional research	8, 9
**Changes in the Physical Environment**	
Doctor’s Office	3, 4, 5, 6, 7, 8, 10, 11
Fewer leisure environments	5, 6
Local Pharmacy	2, 3
Emotional Transformation	
Would feel happy/glad	4, 7, 12
Practice for Change	
Phone reminders	5, 6, 9, 10
Take medication with a partner	7, 8
Seeing their doctor more frequently	2
Taking it with birth control	11
**Changes in the Social Environment**	
Support from partners	1, 2, 3, 4, 6, 9, 11, 12
Support from friends	7, 10
Support from doctor	5
Support from family	1

## Data Availability

The raw dataset supporting the conclusions of this article will be made available by the authors on request.

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
