# Peer review of "Developing a Theory-Based Instrument for Pre-Exposure Prophylaxis (PrEP) Uptake in People of Color Using a Qualitative Approach"

_healthcare, 2024, doi:10.3390/healthcare12161595_

Round 1

Reviewer 1 Report

Comments and Suggestions for Authors

This paper presented a study to create a theory-based instrument that can quantitatively measure the factors infecting the utilization of PrEP among communities of color compared to American whites. Authors proposed to use this instrument to study the underlying reasons for the disparity of PrEP uptake among communities of color compared to white Americans. Based on the interview conducted, the cost of PrEP, lack of protection against other STDs, reducing trust between partners and potential stigma are barriers identified through the interview of 12 participants. 

Overall, the quality of the manuscript is great. The proposed instrument utilizes the findings obtained from the interview and identified and appropriately assigned the scores to each questions, reflecting the factors based on the MTM theory.

Minor issues in the paper:

Line 333-335: The first two proposed instrument questions were identical.

Line 335-336: May need clarifications to distinguish between "will want to" and "intend to" at least for the users

Line 485-491: You may want to discuss about limitation or possible biases given the instrument was developed according to 12 participant and why not include American whites since they may provide other aspects that the communities of color did not mention which may be the factor that causes the disparity.

Line 137-148: It might be helpful to summarize the data in a table. Or at least use a stem-and-leaf plot to show the ages for each participant.

Line 18: remove "as"

Reviewer 2 Report

Comments and Suggestions for Authors

Reviewer 3 Report

Comments and Suggestions for Authors

The authors conducted a qualitative study based on Multi-Theory Model (MTM) concerning Pre-exposure 2 Prophylaxis (PrEP) Uptake in People of Color. The study is written in its all details that is a strong aspect of this study. This paper is good enough for further process of publication; however, there are some issues need to be addressed by the authors in a revised version of the paper:

1-The most important issue is about the small sample size! No matter a qualitative or quantitative study, in clinical-related studies we need a suitable sample size enabling the results for extending to a bigger sample size like a community or distinct cohort. It is highly recommended that the authors mention this issue in the limitation section at least. 

2-In the result section, the details have the potential to be confusing for readers, otherwise, the authors have not designed a table. This study may not need a figure, but it really needs at least a comprehensive table for the main results. Please consider a table in the result section. 

3-The included subjects are not balanced (3 females vs. 9 males) and this can have a major impact on the outputs and conclusions. It is an important factor due to sextual relationships in PrEP. If a study want to replicate the authors' results in a quantitative work, it will be face with a major concern about unbalanced number of male/female. 
